# Enhancing Key Management in LoRaWAN with Permissioned Blockchain

**DOI:** 10.3390/s20113068

**Published:** 2020-05-29

**Authors:** Victor Ribeiro, Raimir Holanda, Alex Ramos, Joel J. P. C. Rodrigues

**Affiliations:** 1University of Fortaleza (UNIFOR), Fortaleza 60811-905, Brazil; raimir@unifor.br; 2Federal Institute of Ceará (IFCE), Fortaleza 60040-531, Brazil; alex.ramos@ifce.edu.br; 3Federal University of Piauí (UFPI), Teresina 64049-550, Brazil; joeljr@ieee.org; 4Instituto de Telecomunicações, 6201-001 Covilhã, Portugal

**Keywords:** Internet of things, low-power wide area network, LoRawan, security, blockchain, hyperledger

## Abstract

Low-Power Wide-Area Network (LPWAN) is one of the enabling technologies of the Internet of Things (IoT), and focuses on providing long distance connectivity for a vast amount of smart devices. Currently, LoRa is one of the leading LPWAN solutions available for public use. In LPWANs, especially in LoRa, security is a major concern due to the resource constraints of the devices, the sensitivity level of the transmitted data, the large amount of connected devices, among other reasons. This paper studies the key management mechanism of LoRaWAN environments. A secure architecture for key management based on smart contracts and permissioned blockchain to enhance security and availability in LoRaWAN networks is proposed. To demonstrate the feasibility of the proposed blockchain-based LoRaWAN architecture, a working prototype has been created using open-source tools and commodity hardware. Performance analysis shows that the prototype presents similar execution time and latency values, when compared to a traditional system, especially for small and medium-sized LoRaWAN networks. We also discuss why the proposed solution can be used in environments with a large number of end-devices.

## 1. Introduction

Low-Power Wide-Area Network (LPWAN) is one of the enabling technologies of the Internet of Things (IoT). An LPWAN is a network designed to allow long-range communications among smart devices, at low bit-rates [1]. This type of network applies to IoT scenarios where low-cost devices need to transmit small messages over long distances (a few kilometers) and transmission delay is tolerable by the application. In LPWANs, end-devices are connected to a gateway node in a direct manner, thus forming a star topology. The simplified design of LPWANs and duty cycling provide a significant reduction in power consumption. Moreover, the great majority of communications in an LPWAN occur from the end-node to the gateway (uplink). This enables end-devices to enter in sleep mode after sending a message, and remain sleeping for long time intervals. Currently, several LPWAN solutions are available, among them LoRa [2] stands out as one of the most promising LPWAN technologies.

In LPWANs, especially for LoRa, security is a major concern. In the literature, several works have exposed the susceptibility of LoRa to attacks, in the phases of key management, network connection, and communications [3,4,5,6]. Although many security improvements have been added to the architecture of LoRa in more recent specifications, much work still needs to be done in this regard.

Blockchain is a disruptive technology that offers many benefits that can be used in IoT scenarios to address several challenges [7,8,9]. Such benefits include decentralization, scalability, security and privacy. Blockchain eliminates the necessity for a central authority, since it works as a distributed database capable of storing every single operation performed by participating parties in a given system. All this is done with extensive use of consensus-based techniques and cryptography, in order to provide important features such as data security, auditability, and anonymity.

In the LPWAN domain, the integration between Blockchain and LoRaWAN is already present in some of the literature [10,11,12,13,14]. The authors of Reference [10] propose maintaining a historical registry of all sensor data gathered from LoRaWAN deployments by making the Network Servers behave as blockchain nodes. In this work, the use of blockchain allows the data to be stored in a distributed and secure fashion. An analogous proposal is made by the authors of Reference [11]; however they also turn the gateway into a blockchain node, not only the Network Servers. Another blockchain-based LoRaWAN infrastructure is proposed in References [12,13]. In this work, blockhain is used to make it possible to share, in a trusted manner, the Network Server from a given provider among different clients responsible for end-devices. Such clients also maintain their own Network Servers. All these studies contribute to the fully decentralized vision of the IoT, however none of them address the key management scheme of the LoRaWAN standard. Such mechanism is extremely relevant to provide security features such as confidentiality, integrity and authentication, but in the current LoRaWAN architecture, it becomes a single point of failure, since it is performed by a single component.

To tackle these issues, a fault-tolerant and secure architecture for key management in LoRaWAN networks based on a permissioned blockchain is proposed in this paper. In the proposed architecture, a smart contract is responsible for key management in the LoRaWAN environment, enabling a secure and distributed storage that ensures high availability for the security keys used in the authentication and communication of end-devices. Some of the main contributions of the proposed architecture are—(1) the creation of a working prototype using only open-source tools; (2) the proposed architecture can be applied to the latest version of LoRaWAN; (3) modifications are not necessary to the LoRaWAN protocol or to the firmware of end-devices.

The structure of this paper is organized as follows. Section 2 discusses related work regarding LoRaWAN-Blockchain integration. Section 3 presents the necessary background knowledge with an overview of the LoRaWAN standard and the fundamentals of permissioned blockchains. Section 4 describes the blockchain-based key management infrastructure proposed and the implementation details of the working prototype. A performance analysis to evaluate the architecture is presented in Section 5. Finally, Section 6 concludes the paper and suggests future research.

## 2. Related Work

Several studies addressing topics of key management in LoRaWAN networks have been published. Most of these studies are primarily focused on, but not limited to, issues such as the generation, exchange and update of encryption keys.

In Reference [15], the authors propose the use of a trusted third party for key management. First, the session keys (NwkSKey and AppSKey) are created by a trusted third party and then transferred to the Network Server and Application Server. Second, the use of timestamps, instead of nonces, is suggested to avoid replay attacks. Third, the authors also propose the derivation of a new AppKey by using a one time password (OTP) with the old root key as an input parameter.

A key management approach based on Ephemeral Diffie-Hellman Over COSE (EDHOC) is presented in Reference [16]. The authors propose the use of a lightweight public key cryptography mechanism to enable end-devices to update session keys in a secure way. EDHOC messages are transmitted as CoAP payloads. In order to transmit IPv6 datagrams over LoRa, the Static Context Header Compression (SCHC) algorithm is used to reduce the size of IPv6 headers.

A key managament scheme for updating root keys in LoRaWAN 1.1 is proposed in Reference [17]. This scheme consists of a two step Key Derivation Function (KDF). The KDF scheme begins with a randomness extraction step, followed by a key expansion step. The Rabbit stream cipher is used as a pseudo-random number generator for the two steps. The authors compare the performance of the proposed scheme with HKDF and AES-CMC-Based-KDF. Performance results show that the proposed scheme presents a high degree of randomness and lower execution times when compared to other schemes.

A session key generation method for LoRaWAN servers, named “Server Session Key Generation” (S2KG), is presented in Reference [18]. In the S2KG method, symmetric and asymmetric encryption are combined to generate session keys for communication between servers in a LoRaWAN network. Elliptic curve cryptography is used to secure communications involving the Join Server, while the AES algorithm is utilized by the remaining servers, for example: communications between network server and application server. The Scyther verification tool was used to evaluate the security of the proposed method. Currently, several researches addressing the integration between IoT and Blockchain are available in the literature [19,20,21,22]. The great majority of them address IoT in a more general context. There are also some works focusing specifically on LoRAWAN.

In References [10,23], a conceptual infrastructure is proposed to add blockchain capabilities to LoRaWAN deployments by design. In this proposal, Network Servers form a peer-to-peer architecture and are enabled to carry out blockchain operations, such as block chaining, transaction verification and hashing. This build-in blockchain solution proposed forms a trusted system which is decentralized and is capable of verifying whether a given transaction has existed at a specific moment in time. One drawback of the proposed solution is that it has not been implemented, which makes it difficult to validate its feasibility. Another issue is that the authors have not described the security aspects of the architecture.

An architecture in which end-devices are connected to a blockchain-based infrastructure is also proposed in Reference [11]. The additional feature of this work is that the gateway also acts as a blockchain node, not only the Network Servers. The authors implement their architecture using Ethereum [24] and real LoRa devices. However, the proposal lacks details about the integration between the blockchain nodes and the Application Servers. In addition, the proposal does not describe the security features of the infrastructure. In Reference [25], a similar solution is presented but designed for a Pollution Monitoring application.

In Reference [12], Durand et al. proposes a decentralized blockchain-based solution for LoRaWANs centered in the roaming feature. When this feature is used, the Network Servers of a provider work as a broker (forwarding server) between end-devices and Network Servers owned by the participant (the entity who is also responsible for the Application Servers). In this architecture, a smart contract is used as a Domain Name Service which allows the Network Servers from a provider to find the participant’s Network Server that is responsible for a given end-device. The strength of this proposal is that third parties are not needed to perform JS resolving. Sharing Join identifiers with partners is not necessary as well. However, the system is implemented based only on the LoRaWAN 1.0 specification. Furthermore, digital signatures over MAC messages are not implemented.

In Reference [13], Durand et al. then improves the aforementioned decentralized blockchain-based LoRaWAN proposal by implementing a new mechanism of digital signatures to ensure the non-repudiation security requirement. This allows to create a totally decentralized system based on smart contracts. It also enables to evaluate whether gateway owners are making fair use of the network.

In Reference [26], the authors proposed a two-factor authentication mechanism based on blockchain to improve the security of the LoRaWAN Join procedure. The additional security layer inserted into the system allows to increase trust on end-devices. The proposed scheme integrates the standard authentication of the join procedure with a blockchain-based authentication, carried out in an isolated blockchain network. A special node, called agent node, is used to mediate the communication between the blockchain and the LoRaWAN nodes (e.g., Network Servers, gateways). The smart contract that performs the second authentication step is stored and executed in the agent node. This solution is implemented using the Ethereum network. The results obtained from the experiments demonstrate the efficiency of the solution, with respect to latency and throughput. Nonetheless, a considerable amount of delay is introduced in the first joint request, due to the mining procedure executed in the blockchain network. Moreover, since end-devices need to handle specific data to make the two factor authentication work, it is necessary to modify the firmware of the end-devices in order for the proposed solution to work, increasing the interdependency between the LoRaWAN and the blockchain networks.

In contrast to the works just described, the work proposed in the present paper focuses on the key management mechanism of LoRAWAN, interacting only with the JS, enforcing system availability, security and auditability. Moreover, this paper goes further than works which have only proposed a conceptual architecture and provides a working implementation for the proposed architecture. This implementation allows to demonstrate the feasibility of the proposed solution and to handle technical issues that are not identified in conceptual papers. In addition, the implemented prototype is compatible with both LoRaWAN specifications (1.0 and 1.1).

## 3. Background Knowledge

This section presents an overview of LoRaWAN and permissioned blockchains. First, the main characteristics of LoRa and LoRaWAN are explained. Second, the security features of LoRaWAN are described to understand how confidentiality, integrity and authentication are provided in a LoRaWAN network. Finally, this section ends with a brief explanation of permissioned blockchains and Hyperledger Fabric.

### 3.1. LoRa and LoRaWAN

LoRa is an LPWAN technology based on spread spectrum modulation techniques that provide long-range wireless communication for low power and low cost end-devices [1,27]. The LoRa’s physical layer, developed by Semtech, is based on Chirp Spread Spectrum (CSS) and uses different unlicensed frequency bands: 902-928MHz (US), 863-870MHz (EU), 433MHz (EU), and 779-787MHz (CN).

The MAC layer of LoRA, standardized by the LoRa Alliance, is referred to as LoRaWAN which has an open specification (currently at version 1.1 [28]). The LoRAWAN architecture is based on a star-of-stars topology as presented in Figure 1. In this architecture, the end-devices (e.g., sensors nodes) communicate directly with gateways using the LoRa PHY wireless modulation. The objective of gateway nodes is to forward messages from end-devices to the Network Server (NS). The NS stays at the core of the LoRaWAN topology performing several activities like frame authentication, packet routing and roaming. In turn, the Application Server (AS) is responsible for providing high-level services for end-users by using the uplink messages received from end-devices. To gain access to the network, end-devices need to authenticate first. The node responsible for this task is the Join Server (JS). The JS conducts other security-related tasks such as authentication procedures and the storage of encryption keys.

The LoRaWAN specification defines three classes of end-devices: Classes A, B and C. Each of the these classes has distinct trade-offs between energy consumption and latency:Class A end-devices are battery-operated and may require optimization of battery life. They offer bi-directional communication, in which each uplink transmission is followed by two downlink receive windows.Class B end-devices are those that may require additional downlink slots. To schedule extra downlink windows, a synchronization beacon is transmitted by a gateway to the end-device.Class C end-devices do not have power restrictions (constant power available). Moreover, they have extended receive windows.

### 3.2. LoRaWAN Security

LoRaWAN has several mechanisms to provide security for the whole network. Data encryption is used in two different layers—network and application [29]. Each end-device is configured with one key for the network layer (NwkKey) and another key for the application layer (AppKey). These two keys are referred to as root keys. LoRaWAN uses the Advanced Encryption Standard (AES) encryption algorithm to provide data confidentiality for both the MAC layer and the application layer. AES is also used to ensure frame integrity, by using Message Integrity Codes (MIC).

As previously stated, to join a LoRaWAN network, end-devices need to authenticate first. LoRaWAN offers two authentication procedures—Over-the-Air Activation (OTAA) or Activation By Personalization (ABP). OTAA is a handshake procedure in which the JS and end-devices exchange Join-Request and Join-Accept messages, containing nonce values, that are used to derive new session keys. Once the OTAA procedure is completed, the session keys are used to secure all communications between the authenticated end-device and the LoRAWAN network. LoRaWAN 1.1 features a Rejoin procedure in which backend servers (NS and JS) are able to create a new session context for the end-device. The Rejoin procedure may be used to change radio parameters or reset session keys. This procedure is always initiated by the end-device and has three types of Rejoin-Request messages—0, 1 and 2. Only messages of type 1 involve communicating with the JS, while the remaining types (0 and 2) are sent solely to the NS.

On the other hand, in the ABP procedure all encryption keys necessary for communication are installed in end-devices before deployment. Hence, there is no exchange of Join-Request and Joint-Accept messages. The ABP procedure is a much simpler activation method, however it offers less security, since the encryption keys of an end-device will never change during its whole lifetime.

### 3.3. Permissioned Blockchains and Hyperledger Fabric

Blockchain is a revolutionary technology built on top of a peer-to-peer (P2P) network infrastructure that implements a public distributed ledger where transactions are recorded as an immutable chain of blocks [30]. In other words, a blockchain can be described as a distributed database that stores all digital operations that participating entities execute in a given system. The need for a third party to intermediate a transaction between two participating entities is eliminated with the use of a blockchain due to its decentralized design and the strong cryptographic primitives it is based on. Furthermore, a consensus algorithm is used by the participants to verify each transaction registered in the system. This ensures that the public ledger is tamper-proof. Blockchain also provides auditability for systems, since the information added to the chain can never be altered or erased [30,31].

There are two types of blockchain—public and permissioned [30]. They basically differ from each other in terms of performance, consensus algorithm, and read permission. In public blockchains, the elevated amount of participating nodes and validations result in decreased throughput and increased latency [32]. Also, any node in the world can make part of the consensus process in public blockchains and all transactions are visible to anyone else. On the other hand, in permissioned blockchains, the smaller number of validations and the limited amount of nodes participating in the consensus process result in much faster transactions. Moreover, only authenticated nodes can participate in the consensus process of permissioned blockchains and only the organization responsible for maintaining the blockchain network can decide whether the transactions are restricted or visible to the public.

Hyperledger Fabric is an open-source permissioned blockchain platform maintained by the Linux Foundation [33]. Hyperledger Fabric is highly modular as it allows pluggable components, such as consesus algorithms and membership services. Hyperledger Fabric supports smart contracts which are self-executing code used to digitally enforce agreements and other rules in a blockchain environment [34]. As opposed to public blockchains, such as Bitcoin and Ethereum, Hyperledger Fabric does not rely on a cryptocurrency which results in no transaction costs. Smart contracts are referred to as *chaincode* in Hyperledger Fabric and can be written with general purpose languages, such as Golang and Java. Chaincode stores and retrieves data from the blockchain as key-value pairs. Hyperledger Fabric also allows the creation of network partitions, known as channels, in which separate ledgers are shared between a group of authorized peers. Only the members of a specific channel have access to transactions on that channel. This allows confidential transactions to be carried out by a group of trusted members.

A typical Hyperledger Fabric network is composed of the following components:**Peer**: Peers are essential components of Hyperledger Fabric as they are responsible for storing copies of ledgers and chaincodes. Peers can be divided in two types: endorsers and committers. Endorsing peers validate and endorse transaction proposals made by clients. Commiting peers verify if a transaction is endorsed and commit it to the blockchain.**Orderer**: An orderer receives endorsed transactions from clients, orders the new transactions into a block and sends it to the commiting peers. Orderers also performs basic access control by enforcing read and write restrictions for the ledgers.**Membership Service Provider (MSP)**: All components of Hyperledger Fabric must prove their identity in order to participate in the network. The MSP is responsible for defining how the identity of each component is validated. A Certificate Authority (CA) is used by the MSP to create and revoke certificates in the blockchain network.**Client**: Clients are authenticated applications which are able to send transaction proposals to the blockchain network.

Figure 2 shows the transaction flow in a Hyperledger Fabric network. First, a transaction proposal is created by a client and sent to a group of endorsing peers. Second, the proposal is validated and signed by the endorsing peers. Third, the signed transaction proposal is delivered to a orderer which will create a new block of transactions. Finally, the block will be verified by the commiting peers, distributed to all peers on the channel and added to the blockchain.

Due to its attractive features, a permissioned blockchain is a strong candidate to improve the LoRaWAN architecture. Particularly, the authentication and key management process conducted by the JS needs to be always available, fast, auditable and secure. In order to provide all these features, the present work improves the JS from the LoRaWAN architecture by integrating it with Hyperledger Fabric to store the cryptographic keys used in the authentication of end-devices. More precisely, the JS takes advantage of the following features: (1) decentralization to provide high availability; (2) security, ensured by the use of digital signatures and hashing to enable integrity verification of all transactions in the network, and to validate the ownership of data, allowing to easily identify any tampering attempts; (3) confidentiality, as transactions are restricted to a group of trusted peers; (4) immutability of the ledger, which stores, in a secure manner, all transactions data (along with a timestamp) as a chain of blocks in order to provide system auditability [30,31].

## 4. Proposed Architecture

In the current LoRaWAN architecture, the JS is a single point of failure, from a security point of view. This is due to the fact that the JS is responsible for handling the OTAA procedure and storing copies of all encryption keys. Therefore, it becomes a important target for attackers. If, for example, the JS suffers a Denial-of-Service attack, the entire LoRaWAN network can be significantly impacted and stop working.

To handle this problem, this paper proposes to modify the LoRaWAN architecture by adding a secure and distributed storage functionality based on blockchain to support the key management procedure. In this new architecture, the encryption keys are stored in a permissioned blockchain infrastructure, instead of being centrally held by the JS.

The main advantages of our proposal are as follows:To increase data availability by means of a distributed storage provided by multiple peers from a permissioned blockchain, thus preventing the single point of failure issue.To provide system auditability by keeping immutable and verifiable records of all OTAA attempts carried out by end-devices and management operations (e.g., creation and updates of root keys).No changes are required to the LoRaWAN protocol because all the modifications are done only to the JS. All changes occur in a transparent manner to the end-devices. Therefore, there is no need for firmware updates to end-devices deployed on a network.

Key update and key storage are among the main security challenges of key management schemes. Key update ensures that keys are only updated by secure entities, in a secure fashion. Key storage ensures that keys are held in a secure location that guarantees their confidentiality, integrity and availability. While the current architecture of the LoRaWAN JS satisfies the key update feature, it falls short with respect to the key storage feature, especially regarding availability.

Availability and fault tolerance are built-in features provided by the distributed design of blockchain. In permissioned blockchain networks, a group of authenticated peers is responsible for maintaining the distributed ledger, which is replicated in each of these peers. Therefore, even if some peers become unavailable, the data can still be obtained from the remaining peers.

In the proposed architecture, all of the key management tasks are conducted by a smart contract. First, the transactions created by the smart contract are validated. Then, the validated data is inserted into a new block. Later, this block is appended to the blockchain. When this process is concluded, each block will contain immutable records of the key management operations executed by the nodes in the LoRaWAN network. By allowing the verification of all authentication records, the proposed architecture enables system auditability, thus contributing to enhance overall system security. For instance, in this system, an auditor is able to check when a new end-device has gained access to the network or when an existing pair of root keys has been updated.

### 4.1. Network Topology

Figure 3 shows the topology of the architecture proposed in this paper. While the JS still exists in this architecture, it works as a client in the Hyperledger Fabric network, which will be the only responsible for storing the key management data. Since sensitive information will be held in a distributed ledger, data confidentiality is an essential requirement. Thus, to compose the proposed architecture, a permissioned blockchain has been selected. Such type of blockchain provides the following features:Greater authentication and confidentiality degree, in contrast to public blockchains, in which information is held in a public global ledger;Better performance, resulting from the application of more lightweight consensus algorithms (e.g., Practical Byzantine Fault Tolerance—PBFT [35]).

In Hyperledger Fabric, all participating nodes are known and authenticated by means of a Public Key Infrastructure (PKI). It is possible to verify the identity of a given peer by applying its public key and a certificate that is signed by a MSP. The Transport Layer Security (TLS) protocol is used to ensure message confidentiality in the communications between participating nodes. A smart contract is installed and executed in multiple peers of the blockchain network.

### 4.2. Smart Contract Description

Only the JS should be able to access the smart contract (basic access control rules are enforced at the chaincode level). Table 2 presents the *DeviceKeys* struct defined in ChirpStack. By means of this struct, ChirpStack stores and manipulates root keys in the JS. In order to facilitate the integration with the JS, the smart contract was implemented to store and retrieve *DeviceKeys* structs from the blockchain. The *DeviceKeys* struct contains: a pair of timestamps (creation and update), an unique identifier (DevEUI) stored as an 8-byte array, the root keys (NwkKey and AppKey) stored as 16-byte arrays, and the Join Nonce value (used by the JS to create the Join-Accept message).

**Listing 1 sensors-20-03068-t002:** DeviceKeys data structure.

type DeviceKeys struct {	
	CreatedAt	time . Time
	UpdatedAt	time . Time
	DevEUI	[8]byte
	NwkKey	[16]byte
	AppKey	[16]byte
	JoinNonce	int
}

An essential set of operations is implemented by the smart contract in order to support the key management procedure in the LoRaWAN network, as described below:**RegisterDeviceKeys**: This operation should be called by a network administrator during the registration of a new end-device. This function allows to store a new pair of root keys in the blockchain. It receives the identifier of a given end-device (DevEUI) and a *DeviceKeys* struct as input parameters. In order to not store cleartext data in the blockchain, all keys are encrypted using a Key Encryption Key (KEK) before calling this function.**GetDeviceKeys**: This function receives a DevEUI as a input parameter and returns a *DeviceKeys* struct from the blockchain. After invoking this operation, the JS decrypts the keys using KEK. The JS calls this function, during an OTAA procedure, to derive new session keys and create the contents of the Join-Accept message.**UpdateDeviceKeys**: This function allows to modify the *DeviceKeys* struct of an end-device. It can be invoked, for example, when an end-device has been compromised. This function receives a DevEUI and a *DeviceKeys* struct as input parameters. As a result, the *UpdatedAt* field is updated and the new root keys are stored in the blockchain.**RevokeDeviceKeys**: This function should be called by a network administrator whenever an end-device leaves the network. This function takes a DevEUI as a input parameter and clears the contents of the encryption keys from the corresponding end-device (it does so by setting all bits to zero).

### 4.3. System Workflow

Every time a new end-device is deployed to the LPWAN, its root keys must be stored in Hyperledger Fabric. In order to avoid the exposure of sensitive data in transactions, the root keys must be encrypted before calling the smart contract. First, the contents of NwkKey and AppKey are encrypted using a KEK. Second, the JS calls the *RegisterDeviceKeys* function from the smart contract passing DevEUI and the encrypted keys as input parameters. Then, the encrypted data is securely stored in the Hyperledger Fabric blockchain.

During the OTAA procedure, the JS must communicate with the smart contract in order to retrieve the root keys of an end-device from the blockchain and derive new session keys. The message flow that occurs throughout the OTAA procedure in the proposed LoRaWAN architecture is presented in Figure 4. The following steps describe the process performed for OTAA:In the fist step, the end-device initiates the OTAA procedure by sending a Join-Request message to the NS. This message is transmitted in clear text and is composed of three fields: JoinEUI, DevEUI, and DevNonce. The DevEUI field is an unique identifier value for the communicating end-device. The DevNonce field is a simple counter value of 2 bytes that is sent to be used in the creation of the session keys in a posterior moment.The Join-Request message is received and validated by the NS, who sends to the JS a JoinReq message containing the payload of the Join-Request message.When the JS receives the JoinReq message, it calls the *GetDeviceKeys* function from the smart contract located in the Hyperledger Fabric blockchain. At this moment, the smart contract searches the distributed ledger for the encrypted data associated to the DevEUI value. Once the result is returned, the JS decrypts the data with KEK and derives the new session keys using the decrypted root keys. Then, a JoinAns message, containing the derived session keys, is transmitted to the NS.The NS sends to the end-device a Join-Accept message (obtained from the payload of the JoinAns message received from the JS).Once the end-device receives a Join-Accept from the NS, the new session keys are derived. At this moment, the end-device and the JS share the same session keys and the end-device is ready to send data packets to the AS.When the first data packet from the end-device is received by the NS, the application session key (AppSKey) is delivered to the AS.The data packet from the end-device is then forwarded from the NS to the AS. Finally, the AS decrypts the payload in the data packet by using the AppSKey.This is an optional step that will only occur if the NS fails to deliver the AppSKey to the AS. In this case, the AS will request the AppSKey directly to the JS.

The proposed architecture could also be applied to Rejoin procedures since only Rejoin-Request messages of type 1 are sent to the JS. Because LoRaWAN uses different counters for Rejoin-Requests, it would be necessary to store new values in the blockchain. The Rejoin procedure was not tested during this research and it is left as a future work.

### 4.4. Prototype Implementation

The working prototype was implemented using only public open-source tools and commodity hardware. The ChirpStack project [36] (previously known as LoRaServer) was used to create the LoRaWAN infrastructure. ChirpStack features open-source implementations of various LoRaWAN components.

LoPy development boards, from Pycom [37], were used as end-devices. These boards are equipped with a Semtech LoRa transceiver SX1276 and an Espressif ESP32 chipset. During the tests, some LoPy boards were configured as class A end-devices and other LoPy boards as Nano Gateways (1-channel LoRaWAN gateway).

With respect to blockchain, Hyperledger Fabric was used to build the permissioned blockchain network and the smart contract which handles the key management functions. Hyperledger Fabric provides containerized Docker images for different blockchain network components (e.g., peers, orders and certificate authorities).

The ChirpStack project implements the JS as a web server with a RESTful API (JSON and gRPC). The encryption keys were stored in a centralized relational database in the original setup of ChirpStack. The code from the ChirpStack JS was modified in order to call the smart contract and store the root keys in the blockchain. To do so, the official Golang SDK from Hyperledger Fabric was used.

## 5. Results and Discussion

This section presents a security analysis and performance evaluation of the proposed architecture. The performance results are compared with the original ChirpStack setup.

### 5.1. Security Analysis

A secure architecture should guarantee a set of fundamental security requirements. The proposed architecture aims to integrate Hyperledger Fabric with the JS. The security of the remaining LoRaWAN elements (e.g., network server and application server) is out of scope in this paper but a formal security analysis of LoRaWAN can be found in Reference [38]. The following items describe the solutions used by the proposed architecture to ensure key principles of security:**Confidentiality**: The confidentiality principle mandates that data should only be accessed by authorized peers. As a permissioned blockchain platform, Hyperledger Fabric supports confidential transactions by limiting blockchain access only to trusted peers. All communications between peers in the Hyperledger Fabric network are secured by using the TLS protocol. Additionally, the proposed architecture encrypts all data with a KEK before sending it to the permissioned blockchain. This works as an extra layer of security to prevent information leakage in transactions. For instance, even if an authorized peer is compromised, an attacker would only get encrypted data from the blockchain.**Integrity**: Integrity means that data should not be modified in an unauthorized manner. The hashing mechanisms used by Hyperledger Fabric ensure that the integrity of transactions and data stored in the blockchain can be verified in a secure manner. Any attempt to modify the data on the blockchain would result in invalid hash values (different from the values found in the blocks).**Availability**: The availability principle states that a service or data must be available whenever necessary. Hyperledger Fabric offers availability by storing copies of all records in multiple peers. The decentralized peer-to-peer design of permissioned blockchains enables high availability of data. Since Hyperledger Fabric uses the PBFT consensus algorithm, it tolerates f < n/3 faulty nodes [39]. Therefore, the proposed architecture is resilient to Denial-of-Service attacks.**Authentication**: As a permissioned blockchain, Hyperledger Fabric allows access only to authorized peers. Public-key cryptography is used by a membership service in Hyperledger Fabric to identify peers through certificates. All trusted peers have valid certificates signed by a CA. This allows mutual authentication between peers in the proposed architecture.**Non-repudiation**: Non-repudiation is the capacity to ensure the authorship of a certain action. Hyperledger Fabric ensures non-repudiation by means of digital signatures. All transactions are signed with the user’s private key. Therefore, the authenticity of a transaction can be verified using the user’s public key.**Authorization**: Access control rules were implemented at the chaincode level in the proposed architecture. Through certificates signed by the CA, the smart contract is able to verify the identify of a user and check their access permissions. Access to the smart contract functions is only allowed for the JS.

### 5.2. Performance Analysis

The performance results obtained by the proposed architecture were compared with the original ChirpStack setup (which uses a traditional relational database). Due to our limited number of available LoPy boards, a Golang script was created to simulate a large number of simultaneous requests to the JS. This script creates Join-Request messages, sends them to the JS API and collects some statistics. Different threads are created to handle each Join-Request. In other words, each thread simulates the behavior of a given LoPy node. Each thread sends a Join-Request and waits for the Join-Accept. If, instead of the Join-Accept, the thread receives an error response, it re-sends the Join-Request. This process is repeated until a Join-Accepted is received by the thread. In order to evaluate the performance of the proposed architecture and the original ChirpStack, data from the OTAA procedure was collected as follows:**OTAA start time**: Unix time when a Join-Request message was sent to the JS.**OTAA finish time**: Unix time when a successful Join-Accept message was received from the JS.

The two architectures were evaluated under different scenarios of OTAA procedure workloads: 10, 50, 100, 250, 500, 750, 1000, 2500 and 5000 simultaneous requests. Each scenario was executed 10 times. The following metrics were chosen based on Reference [40] and used to compare the performance of the two architectures:**Execution time**: For each set of requests, execution time was calculated as the amount of time used to process all OTAA procedures (Difference between the maximum OTAA finish time and the minimum OTAA start time).**Latency**: For each OTAA procedure, latency is the difference between the OTAA finish time and the OTAA start time. The average latency was calculated as the average of latency of all requests in a set.**Throughput**: For each set of requests, throughput is defined as the total of successful OTAA requests divided by the execution time.

A summary of the experiment settings is provided in Table 1. All the performance tests were conducted using a Ubuntu 19.04 desktop machine with Intel(R) Core(TM) i7-7500U CPU @ 2.70GHz and 16 GB of RAM. Virtual machines were created using VirtualBox for the two environments: LoRaWAN network with ChirpStack and permissioned blockchain with Hyperledger Fabric. Each guest OS was running Ubuntu 16.04 with 4 GB of RAM.

Figure 5 shows 95% confidence intervals for the chosen metrics and the the varying workloads. A comparison for the total execution time between the proposed solution and the original ChirpStack setup is presented in Figure 5a. For up to 1000 requests, the original ChirpStack setup is able to process all OTAA requests faster than the proposed architecture. But once it reaches 2500 requests, the execution times of the original ChirpStack setup increase rapidly. This happens because the original JS starts dropping Join-Request messages once it receives a large number of simultaneous requests (e.g., 700 requests). Therefore, the original JS takes more time to process all OTAA requests. The proposed solution is able to process all OTAA requests in a linear fashion. For the set with 5000 requests, the average execution time of the proposed architecture is 73.04 s and 111.29 s for the original ChirpStack setup (a difference of 38.25 s).

The comparison of the average latency is presented in Figure 5b. As the previous comparison, the two solutions demonstrate close values while handling up to 100 requests. The original setup of ChirpStack shows smaller latency values than the proposed architecture. However, the latency difference decreases as the number of requests increases. For the set with 5000 requests, the average latency of the proposed architecture is 59.24 s and 55.29 s for the original ChirpStack setup (a difference of 3.95 s). The close latency values show that the performance of the original ChirpStack setup decreases in large networks.

Figure 5c shows the comparison of the throughput values. The proposed solution shows a small slope up to 250 requests, but once it reaches 500 requests, the throughput values stabilize around 70 requests per second. This is in accordance with the linear behavior of the latency and execution time results, and is due to the Hyperledger’s intelligent requests management, which holds the requests in the server until they are able to be further processed in the blockchain, thus only responding when the request has been successfully processed. In turn, the original ChirpStack setup shows a greater slope up to 100 requests, reaching a maximum of 195.09 requests per second, but the throughput starts to rapidly decline as the number of simultaneous requests increases. This happens due to the increasing number of errors generated by the original JS (as explained previously), that is, unlike Hyperledger, the original JS returns error responses when its relational database is overloaded. The throughput values of the proposed solution start to exceed the values of the original ChirpStack setup once a large number of requests is made (e.g., 3500 requests). Therefore, the throughput of the original ChirpStack setup continuously degrades while the proposed architecture remains stable.

Based on the values obtained from the performance evaluations, it can be concluded that the proposed architecture shows acceptable results when compared to the original setup of ChirpStack. For small and medium size LoRaWAN networks (up to 250 end-devices), the use of a permissioned blockchain presents performance results close to a traditional database. The delays seen in the proposed architecture, created by common blockchain operations, are acceptable even for large LoRaWAN networks since end-devices are expected to perform the OTAA procedure only once a day. The advantages of the proposed solution stand out in scenarios with a large number of end-devices. The performance of the original ChirpStack setup significantly decreases in large environments while the proposed architecture shows a stable performance. Therefore, we conclude that due to scalability issues, the original ChirpStack setup is not suitable for large LoRaWAN networks and the distributed design of permissioned blockchains becomes a major advantage.

## 6. Conclusions and Future Works

This paper presents a secure architecture for the key management mechanism in LoRaWAN networks based on a permissioned blockchain network. The proposed solution makes use of confidential transactions and takes advantage of the decentralized design of blockchains to solve the central point of failure problem of the JS. In addition, due to the blockchain features, all OTAA Join requests executed in the network are recorded in an verifiable and immutable way by the proposed architecture. Finally, to handle the encryption keys of all end-devices, a smart contract was implemented in the architecture.

In order to validate the feasibility of the proposed architecture, a working prototype has been implemented using open-source tools. Hyperledger Fabric was used to build the permissioned blockchain network. Modifications were included in the ChirpStack project in order to enable the communication between the JS and the smart contract.

To evaluate the proposed implementation, a performance analysis was carried out in which different workloads were tested. According to the results obtained, the prototype shows a performance similar to the original ChirpStack configuration for small and medium-sized LoRaWAN networks. The proposed architecture can also be applied to large environments, given that end-devices are expected to perform the OTAA procedure only once a day.

As future work, new functionalities will be added to the key management smart contract, by enabling it to handle OTAA requests directly. The use of multiple ledgers for authentication and application data in LoRaWAN will also be evaluated. Moreover, performance evaluations of the implemented prototype will be done using real hardware. 

## Figures and Tables

**Figure 1 sensors-20-03068-f001:**
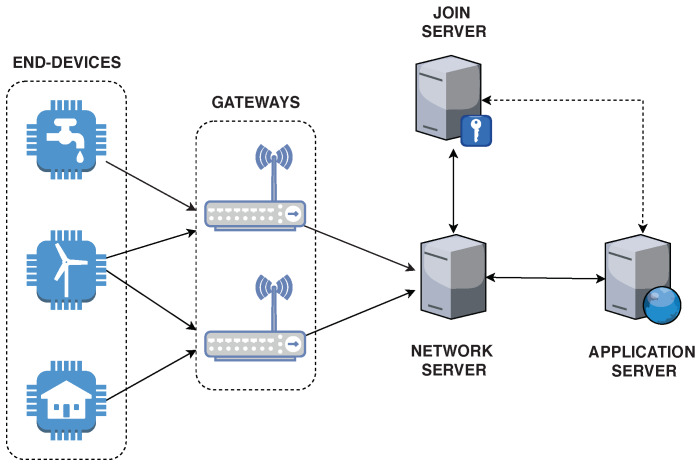
Architecture of a LoRaWAN Network.

**Figure 2 sensors-20-03068-f002:**
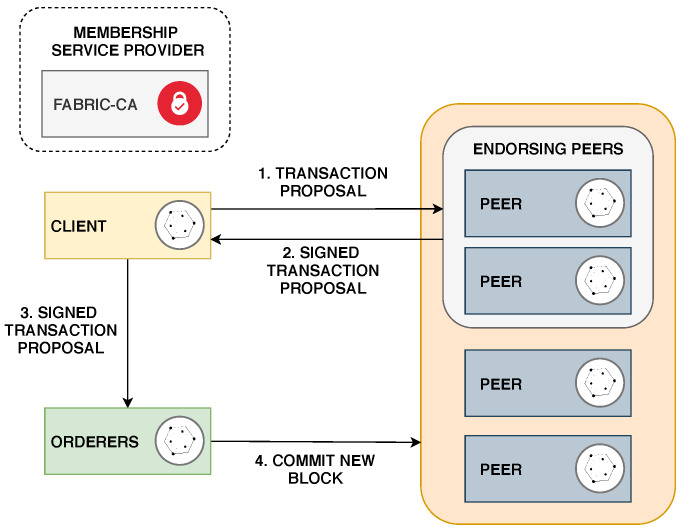
Network architecture and transaction flow in Hyperledger Fabric.

**Figure 3 sensors-20-03068-f003:**
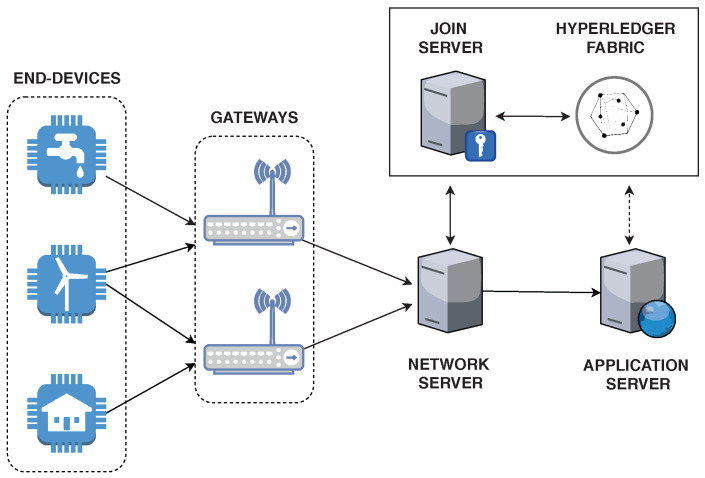
Proposed blockchain-based architecture.

**Figure 4 sensors-20-03068-f004:**
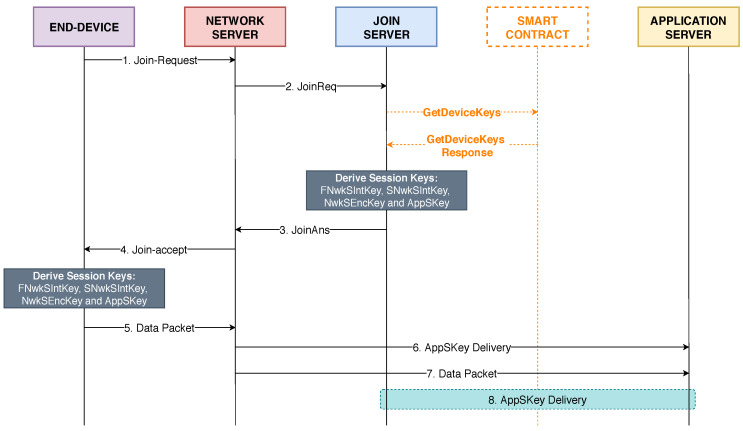
End-device authentication and data transmission in the proposed architecture.

**Figure 5 sensors-20-03068-f005:**
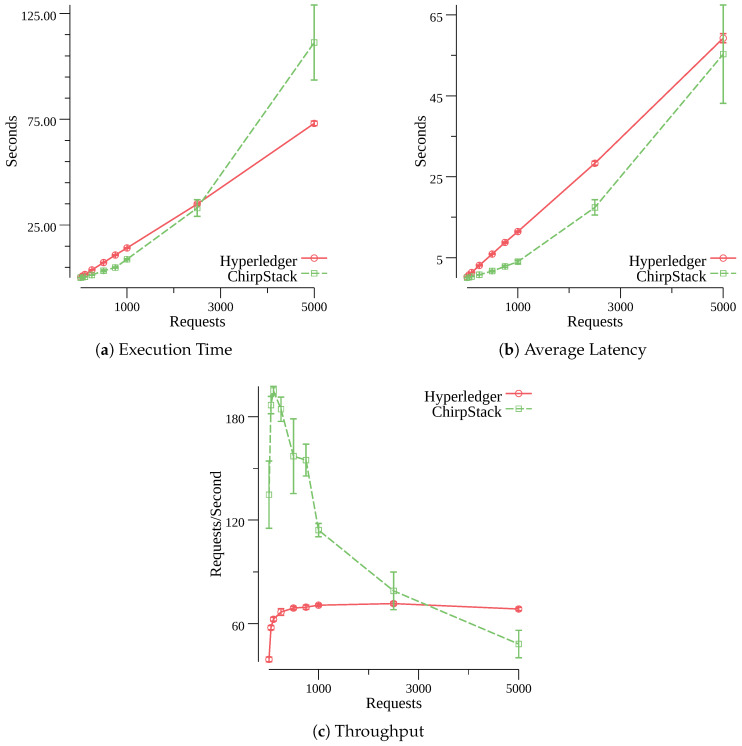
Performance comparison between the proposed architecture (Hyperledger Fabric) and the original ChirpStack setup with different workloads for the OTAA procedure.

**Table 1 sensors-20-03068-t001:** Performance Analysis Settings.

Item	Value
**Number of requests (workload)**	50, 100, 250, 500, 750, 1000, 2500 and 5000
**Number of executions per workload**	10 times
**Evaluated metrics**	Execution time, Latency, Throughput
**Confidence Interval**	95%
**Hyperleger Fabric**	v1.4
**ChirpStack**	v3.3.0

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
