# Peer review of "Enhancing Key Management in LoRaWAN with Permissioned Blockchain"

_sensors, 2020, doi:10.3390/s20113068_

Round 1

Reviewer 1 Report

In general, the paper is appropriate, and addressing a relevant topic related to LoRaWAN security. However, there are several aspects that could be improved:

  1. Even if authors describe the advantages of blockchain and some of the issues regarding LoraWAN security (specially focused on the fact that the JS is a single point of failure), it is not clear which challenges are covered by the use of blockchain regarding key management aspects.
  2. Indeed, authors should include some references focused on key management aspects, such as:
  • Sanchez-Iborra, R., Sánchez-Gómez, J., Pérez, S., Fernández, P. J., Santa, J., Hernández-Ramos, J. L., & Skarmeta, A. F. (2018). Enhancing lorawan security through a lightweight and authenticated key management approach. Sensors18(6), 1833.
  • Han, J., & Wang, J. (2018). An enhanced key management scheme for LoRaWAN. Cryptography2(4), 34.
  • Naoui, S., Elhdhili, M. E., & Saidane, L. A. (2017, October). Trusted third party based key management for enhancing LoRaWAN security. In 2017 IEEE/ACS 14th International Conference on Computer Systems and Applications (AICCSA) (pp. 1306-1313). IEEE.
  • Tsai, K. L., Leu, F. Y., Hung, L. L., & Ko, C. Y. (2020). Secure Session Key Generation Method for LoRaWAN Servers. IEEE Access8, 54631-54640.

and to describe the relationship with such works

3. Authors could extend a bit the part of the implementation (e.g., which version of Hyperledger Fabric are you using?), and the reasons to have a constant throughput in Figure 5c

4. Taking into account the latest version of LoRaWAN (1.1), one would expect to have some references to the Rejoin mechanisms, and how they are related with the proposed approach.

Author Response

- Item 01: "Even if authors describe the advantages of blockchain and some of the issues regarding LoraWAN security (specially focused on the fact that the JS is a single point of failure), it is not clear which challenges are covered by the use of blockchain regarding key management aspects."

Answer to item 01: Key update and key storage are among the main security challenges of key management schemes. Key update ensures that keys are only updated by secure entities, in a secure fashion. Key storage ensures that keys are held in a secure location that guarantees their confidentiality, integrity and availability. While the current architecture of LoRaWAN JS satisfies the key update feature, it falls short with respect to the key storage feature, especially regarding availability (text added to section 4).

- Item 02: "Authors could extend a bit the part of  the implementation (e.g., which version of Hyperledger Fabric are you  using?)"

Answer to item 02: We added more information about the versions of softwares used on Table 1 (section 5.2).

- Item 03: "Authors could explain the reasons to have a constant throughput in Figure 5c"

Answer to item 03: More details regarding the throughput values were added to section 5.2.

- Item 04: "Taking into account the latest version  of LoRaWAN (1.1), one would expect to have some references to the Rejoin  mechanisms, and how they are related with the proposed approach."

Answer to item 04: The proposed architecture could also be applied to Rejoin procedures since only Rejoin-Request messages of type 1 are sent to the JS. Because LoRaWAN uses different counters for Rejoin-Requests, it would be necessary to store new values in the blockchain. The Rejoin procedure was not tested during this research and it is left as a future work. (Text added to sections 3.2 and 4.3)

- Item 05: "Authors should include some references focused on key management aspects"

Answer to item 05: The 4 articles on key management aspects suggested by the reviewer were added to the latest version.

Reviewer 2 Report

The paper proposes to enhance key management of LoRaWAN by leveraging permissioned Blockchain. The ideas proposed in the paper are interesting however the presentation can be improved as follows:

1) In the literature, there is much good work on LoRaWAN Security. Authors need to view and digest them. To name a few:
- Yang, Xueying, et al. "Security vulnerabilities in LoRaWAN." 2018 IEEE/ACM Third International Conference on Internet-of-Things Design and Implementation (IoTDI). IEEE, 2018.
- Eldefrawy, Mohamed, et al. "Formal security analysis of LoRaWAN." Computer Networks 148 (2019): 328-339.
- Butun, Ismail, Nuno Pereira, and Mikael Gidlund. "Security risk analysis of LoRaWAN and future directions." Future Internet 11.1 (2019): 3.
- Tomasin, Stefano, Simone Zulian, and Lorenzo Vangelista. "Security analysis of lorawan join procedure for internet of things networks." 2017 IEEE Wireless Communications and Networking Conference Workshops (WCNCW). IEEE, 2017.

2) Figures 1 and 3 need to be improved, spell out the names of GW and end-node. If the picture is taken from elsewhere, it requires proper citation.

3) Figure 5 is not readable on A4 paper. Needs to be revised.

3) It is stated in the Conclusions that (L409): "This paper presents a secure architecture for the key management mechanism in LoRaWAN ..." The security analysis of the proposed protocol is completely missing. It needs to be verified with formal or informal analysis techniques/tools.

4) Considering that most of the LoRa applications are non-commercial, important questions to be answered here are:
- Who will be the endorsing peers for the Hyperledger?
- What's the benefit for them for serving as an endorsing peer?

Author Response

- Item 01: " Figures 1 and 3 need to be improved, spell out the names of GW and  end-node. If the picture is taken from elsewhere, it requires proper  citation."

Answer to item 01: Figures 1 and 3 were improved in the latest version.

- Item 02: " Figure 5 is not readable on A4 paper. Needs to be revised."

Answer to item 02: Figure 5 was resized in the latest version.

- Item 03: "It is stated in the Conclusions that (L409): "This paper presents a  secure architecture for the key management mechanism in LoRaWAN ..."  The security analysis of the proposed protocol is completely missing. It  needs to be verified with formal or informal analysis techniques/tools."

Answer to item 03: An informal security analysis was added to the paper (section 5.1).

- Item 04: "Considering that most of the LoRa applications are non-commercial, important questions to be answered here are: Who will be the endorsing peers for the Hyperledger? What's the benefit for them for serving as an endorsing peer?"

Answer to item 04: Hyperledger Fabric does not rely on a cryptocurrency, therefore, there are no transaction costs. We consider this a benefit to the network administrator since it results in less overall costs (text added to section 3.3).

- Item 05: "In the literature, there is much good work on LoRaWAN Security. Authors need to view and digest them".

Answer to item 05: The four articles on LoRaWAN security suggested by the reviewer were added to the latest version.

Reviewer 3 Report

The paper tackles key management mechanism of LoRaWAN networks. To this end, the authors propose a secure architecture for key management based on smart contracts and permissioned blockchain.The authors also run a performance study to evaluate and verify the effectiveness of the proposed solution in comparison to a legacy mechanism in terms of execution time and latency. Scalability issues are also discussed.

Further improvements can be possible in the article by considering the following points:

-Related Work section appears near the end of the article. The authors are encouraged to include this section earlier in the paper.

-Performance analysis settings should be explained better. A table to follow the settings would be helpful for the reader.

-Errors of execution time and latency under 5000 requests is remarkably wide. The authors should elaborate further on this phenomenon.

-Similarly for throughput, the abovementioned phenomenon is opposite; i.e., errors under 1000 requests are wider remarkably. The authors should also elaborate on this issue as well. 

-Hyperledger Fabric was used for blockchain implementation. The authors mention it provides containerized Docker images for different blockchain network components. Is this the only reason to select HyperLeger Fabric? Were any other alternatives were available?

-Scalability discussion could be expanded, probably with respect to the state of the art. Existing scalability studies can be used to make a qualitative comparison between the proposed scheme and the previous studies.

-A few examples from the related work are omitted; the authors may consider including these, which are given below but are not limited to the list:

Schiller et al, Scalable Transport Mechanisms for Blockchain IoT Applications, LCN 2019

Moin et al, Securing IoTs in distributed blockchain: Analysis, requirements and open issues, Elsevier FGCS, Nov 2019

Makhdoom et al, Blockchain's adoption in IoT: The challenges, and a way forward, Elsevier J. of Network and Computer Applicatins, 2019

Mohanta et al, Blockchain technology: A survey on applications and security privacy Challenges, Internet of Things (Elsevier), 2019

Mundt et al, General Security Considerations of LoRaWAN Version 1.1 Infrastructures, ACM MObiwac 18

Bragagnolo et al, Towards scalable blockchain analysis, ACM WETSEB 2018

Author Response

Item 01: " Related Work section appears near the end of the article. The  authors are encouraged to include this section earlier in the paper."

Answer to item 01: Related Works was moved to section 2.

Item 02: " Performance analysis settings should be explained better. A table to follow the settings would be helpful for the reader."

Answer to item 02: A summary of the settings used during performance analysis was added to table 1 (section 5.2)

Item 03: "Errors of execution time and latency under 5000 requests is  remarkably wide. The authors should elaborate further on this  phenomenon. Similarly for throughput, the abovementioned phenomenon is opposite;  i.e., errors under 1000 requests are wider remarkably. The authors  should also elaborate on this issue as well. "

Answer to item 03: More details regarding the throughput values were added to section 5.2.

Item 04: "Hyperledger Fabric was used for blockchain implementation. The  authors mention it provides containerized Docker images for different  blockchain network components. Is this the only reason to select  HyperLeger Fabric? Were any other alternatives were available?"

Answer to item 04: We have some motives for choosing Hyperledger Fabric. First, Hyperledger Fabric is one of the most popular blockchains (along with Ethereum) available for public use. Second, smart contracts are written using Go in Hyperledger (this is favorable for us because of our familiarity with Go). Third, ChirpStack was also developed with Go, which makes it easier for us to integrate with Hyperledger.

Item 05: "Scalability discussion could be expanded, probably with respect to the state of the art. Existing scalability studies can be used to make a qualitative comparison between the proposed scheme and the previous studies."

Answer to item 05: To the best of our knowledge, there is no work specifically analysing the scalability of LoRaWAN's traditional join procedure. Discussion about scalability will be left for future work.

Item 06: "A few examples from the related work are omitted; the authors may consider including these, which are given below but are not limited to the list."

Answer to item 06: All of the articles suggested by the reviewer were added to the latest version.

Reviewer 4 Report

Page 1, Line 27: Exactly what do you mean be LoRa’s ”upper layer counterpart”?

P 2, L 76: You write “…designed by the LoRa Alliance…”, I would rather suggest “…standardized by the LoRa Alliance…”.

P 5, L 164-176: This paragraph makes me a bit confused. It is part of the Background chapter and I assume that it is describing the existing state-of-the-art. Still it is written in future tense, like it is describing something that has not yet been done, easily confusing it with your contributions that are to be presented in this paper.

P 10, L 355-407: This is the first time ever that I have seen related work being presented after the results. I would strongly suggest that the related work (except for the last paragraph where you describe your own contribution) is moved to the end of chapter 2 or possibly as a new chapter 3.

Overall, the paper is quite well written, though there are some issues with the structure. However, I find the content is a bit thin and un-theoretic for a journal paper. Perhaps the test/simulation results could be supported by a theoretical evaluation. It would also be nice with a more thorough comparison to the current state-of-the-art. At the current state I think that it would make nice conference paper.

Author Response

- Item 01: "Page 1, Line 27: Exactly what do you mean be LoRa’s ”upper layer counterpart”?"

Answer to item 01: The text was removed from the latest version.

- Item 02: "P 2, L 76: You write “…designed by the LoRa Alliance…”, I would rather suggest “…standardized by the LoRa Alliance…”."

Answer to item 02: The text was corrected in the latest version.

- Item 03: "P 5, L 164-176: This paragraph makes me a bit confused. It is part of  the Background chapter and I assume that it is describing the existing  state-of-the-art. Still it is written in future tense, like it is  describing something that has not yet been done, easily confusing it  with your contributions that are to be presented in this paper."

Answer to item 03: The text was corrected in the latest version.

- Item 04: " P 10, L 355-407: This is the first time ever that I have seen related  work being presented after the results. I would strongly suggest that  the related work (except for the last paragraph where you describe your  own contribution) is moved to the end of chapter 2 or possibly as a new  chapter 3."

Answer to item 04: Related Works was moved to section 2.

Round 2

Reviewer 2 Report

The authors have improved the presentation and quality of the paper significantly. However, there are some minor typos exist, and corrections/edits need to be fulfilled by the authors:

  • Reference to the Figures are lost: P12L467, P13L482

  • Heading of Section-5 is orphan, please shift it to the next page.

  • Figure-5 needs further attention, the graphs are not aligned and the title of the mid-graph is separated and spans in an odd manner.

  • I think this needs to be removed from the text: P14L527 "to the CRediT taxonomy for the term explanation. Authorship must be limited to those who have contributed substantially to the work reported."

  • Some typos:
    • P12L449 " 1000, 2500 and 5000 simultaneous " --> " 1,000, 2,500, and 5,000 simultaneous"
    • P12L464 Table.1 " 1000, 2500 and 5000 simultaneous " --> " 1,000, 2,500, and 5,000 simultaneous"
    • P14L493 "3500 requests" --> "3,500 requests"

Author Response

- Item 01: "Reference to the Figures are lost: P12L467, P13L482"

Answer to item 01: The references were corrected in the latest version.

- Item 02: "Heading of Section-5 is orphan, please shift it to the next page."

Answer to item 02: The header of section 5 has been moved to the beginning of page 11.

- Item 03: "Figure-5 needs further attention, the graphs are not aligned and the title of the mid-graph is separated and spans in an odd manner."

Answer to item 03: The layout of figure-5 has been improved in the latest version.

- Item 04: "I think this needs to be removed from the text: P14L527 "to the CRediT taxonomy for the term explanation. Authorship must be limited to those who have contributed substantially to the work reported.""

Answer to item 04: The text was removed from the latest version.

- Item 05: "P12L449 " 1000, 2500 and 5000 simultaneous " --> " 1,000, 2,500, and 5,000 simultaneous""

Answer to item 05: The text was corrected in the latest version.

- Item 06: P14L493 "3500 requests" --> "3,500 requests"

Answer to item 06: The text was corrected in the latest version.

Reviewer 4 Report

I find the content to be appropriate for publication, though still a bit thin and un-theoretic for a journal paper, but I leave that decision to the editor.  

Author Response

- Item 01: "I find the content to be appropriate for publication, though still a bit thin and un-theoretic for a journal paper, but I leave that decision to the editor."

Answer to item 01: We thank the Reviewer for his/her appreciation of our work.
